# Iterative fully convolutional neural networks for automatic vertebra segmentation

**Nikolas Lessmann**
Image Sciences Institute
University Medical Center Utrecht

**Bram van Ginneken**
Diagnostic Image Analysis Group
Radboud University Medical Center

**Pim A. de Jong**
Department of Radiology
University Medical Center Utrecht

**Ivana Išgum**
Image Sciences Institute
University Medical Center Utrecht

## Abstract

Precise segmentation of the vertebrae is often required for automatic detection of vertebral abnormalities. This especially enables incidental detection of abnormalities such as compression fractures in images that were acquired for other diagnostic purposes. While many CT and MR scans of the chest and abdomen cover a section of the spine, they often do not cover the entire spine. Additionally, the first and last visible vertebrae are likely only partially included in such scans. In this paper, we therefore approach vertebra segmentation as an instance segmentation problem. A fully convolutional neural network is combined with an instance memory that retains information about already segmented vertebrae. This network iteratively analyzes image patches, using the instance memory to search for and segment the first not yet segmented vertebra. At the same time, each vertebra is classified as completely or partially visible, so that partially visible vertebrae can be excluded from further analyses. We evaluated this method on spine CT scans from a vertebra segmentation challenge and on low-dose chest CT scans. The method achieved an average Dice score of 95.8 % and 92.1 %, respectively, and a mean absolute surface distance of 0.194 mm and 0.344 mm.

## 1   Introduction

Segmentation of the vertebrae is often a prerequisite for automatic analysis of the spine, such as detection of vertebral fractures [1]. Automatic spine analysis is especially relevant in images that were originally not intended for spine imaging, such as CT scans of the chest or abdomen, in which it allows for detection of spinal diseases as an unrequested finding [2]. In these images, only part of the spine is typically visible, and some of the visible vertebrae may be only partially visible. Detection of spinal diseases is often based on analysis of individual vertebrae rather than analysis of the spine as a whole. Accordingly, the automatic segmentation needs to distinguish individual vertebrae and needs to exclude those that are only partially visible in the image. Vertebra segmentation is therefore essentially an instance segmentation problem for which the number of instances (completely visible vertebrae) is not known *a priori*.

Traditionally, spine segmentation has not been approached as an instance segmentation problem, but predominantly as a model-fitting problem using methods such as deformable models or atlas-based segmentation [3–6]. Many recent methods rely on machine learning including deep learning, but often still in combination with model-based strategies. Sekuboyina *et al.* [7] used a simple multi-label 2D fully convolutional neural network (FCN) for segmentation of only the lumbar vertebrae. Korez *et al.* [8] used a convolutional neural network to generate probability maps that indicate the location of

1st Conference on Medical Imaging with Deep Learning (MIDL 2018), Amsterdam, The Netherlands.

vertebral bodies and used these maps to guide a deformable model. Similar ideas were used to only localize vertebrae: Yang *et al.* [9] detected vertebra centroids by using a FCN to obtain a probability map for each vertebra followed by a message passing scheme to extract a plausible set of centroids. Chen *et al.* [10] detected vertebrae with a convolutional neural network trained with a specialized loss term to help distinguish neighboring vertebrae.

In this paper, we propose to segment vertebrae with an iterative instance segmentation approach that detects and segments vertebrae successively. We combine a 3D segmentation FCN with a memory component, which remembers already segmented vertebrae across successive iteration steps, and an instance classifier that identifies completely visible vertebrae. This approach is similar to instance segmentation algorithms based on recurrent networks [11] and the iterative non-recurrent approach by Li *et al.* [12]. However, these algorithms do not generalize well to 3D volumes because they analyze the entire image at once, which is currently not feasible with typical 3D medical image volumes due to hardware limitations. To cope with this problem, we previously proposed a two-stage approach based on two consecutive FCNs, of which the first operates on downsampled images [13]. In this paper, we instead incorporate anatomical information, namely the information that the next instance can be expected in close proximity to the current instance, which enables a search and track strategy based on image patches. This method can analyze images of arbitrary size and field of view and it does not pose any requirements regarding the visibility of certain vertebrae. We evaluated it on a set of dedicated spine CT scans as well as a challenging set of low-dose chest CT scans.

## 2 Method

Our approach consists of three major components that are combined with an iterative inference strategy. The central component is a *segmentation network* that segments voxels of vertebrae from a 3D image patch. To enable this network to segment only voxels belonging to a specific instance rather than all voxels belonging to all visible vertebrae, we augment the network with an *instance memory* that informs the network about already segmented vertebrae. The network uses this information to always segment only the following not yet segmented vertebra. The segmentation process is therefore iterative: Patches along the spine are fed to the segmentation network and once the first vertebra is fully segmented, the instance memory is updated. This triggers the network to focus on the next vertebra in the next iteration. The third component is a *classification network* that is added to the network to distinguish between completely visible and partially visible vertebrae. The full network architecture is illustrated in Figure 1.

**Segmentation network**

The segmentation component of the network is a FCN that predicts a binary label for each voxel in a patch of the analyzed image. This label indicates whether the voxel belongs to the current instance or not. We used a patch size of $128 \times 128 \times 128$ voxels, which is large enough to cover an entire vertebra. To ensure that all patches have the same resolution, even though the analyzed images may have different resolutions, we resample all input images to an isotropic resolution of $1 \, \text{mm} \times 1 \, \text{mm} \times 1 \, \text{mm}$ prior to the segmentation. The obtained segmentation masks are resampled back to the original image resolution using nearest neighbor interpolation.

The architecture of the segmentation network is inspired by the U-net architecture [14], i.e., consists of a contraction and an expansion path with intermediate skip connections. We use a constant number of filters in all layers and added batch normalization as well as additional padding before all convolutional layers to obtain segmentation masks of the same size as the image patches. The segmentation masks are additionally refined by removing voxels below $200 \, \text{HU}$ from the surface of each vertebra.

**Instance memory**

In the proposed iterative segmentation scheme, detailed in the following section, the network segments one vertebra after the other, one at a time. The purpose of the instance memory is to remind the network of vertebrae that were already segmented in previous iterations so that it can target the next not yet segmented vertebra. This memory is a binary flag for each voxel of the input image that indicates whether the voxel has been labeled as vertebra in any of the previous iterations. Together

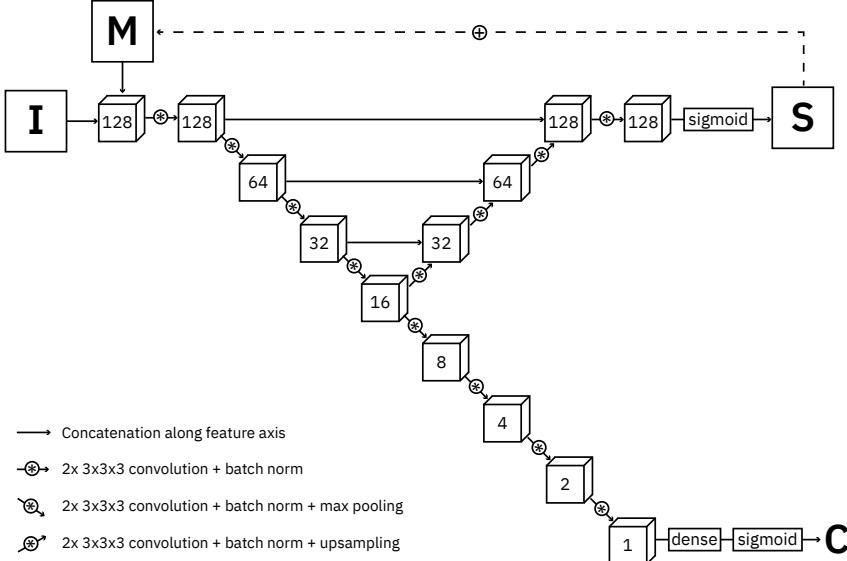

Figure 1: Schematic drawing of the network architecture. **I** denotes the *input patch*, **M** the *instance memory*, **S** the *segmentation* (probability map), and **C** the *classification* (single probability value for complete visibility). Cubes represent 3D feature maps with 64 channels, with exception of the first cube after I/M, which has two channels, and the cube before S, which has only one channel. The number on each cube indicates the size of the feature map (e.g., $128 \times 128 \times 128$ voxels).

with an image patch, the network receives a corresponding memory patch as input. These are fed to the network as a two-channel input volume.

**Iterative instance segmentation**

The iterative segmentation process is illustrated in Figure 2.[1] This process follows either a top-down or bottom-up scheme, i.e., the vertebrae are not segmented in random order but successively from top to bottom, or vice versa. The network learns to infer from the memory patch which vertebra to segment in the current patch. If the memory is empty, i.e., no vertebra has been detected yet, the network segments the top-most or bottom-most vertebra that is visible in the patch, depending on the chosen direction of traversal. Otherwise, the network segments the first following not yet segmented vertebra.

The patch size is chosen large enough to always contain part of the following vertebra when a vertebra is in the center of the patch. This enables utilizing prior knowledge about the spatial arrangement of the individual instances to move from vertebra to vertebra. The FCN iteratively analyzes a single patch centered at $x_t$, where $t$ denotes the iteration step. Initially, the patch is moved over the image in a sliding window fashion with constant step size $\Delta x$, searching for the top-most vertebra when using a top-down approach, or the bottom-most vertebra when using a bottom-up approach. As soon as the network detects a large enough fragment of vertebral bone, at least $n_{\min} = 1000$ voxels $\hat{=} \, 10 \, \text{mm}^3$ in our experiments, the patch is moved toward this fragment. The center of the bounding box of the detected fragment, referred to as $b_t$, becomes the center of the next patch:

$$x_{t+1} = \begin{cases} x_t + \Delta x & \text{if } n_t < n_{\min} \\ [\, b_t \,] & \text{otherwise} \end{cases}$$

Even if initially only a small part of the vertebra is visible and detected in the patch, centering the following patch at the detected fragment ensures that a larger part of the vertebra becomes visible in the next iteration. Eventually, the entire vertebra becomes visible, in which case the patch

---

[1] An animation of the iterative segmentation process is available at http://qia.isi.uu.nl/vertebra-segmentation

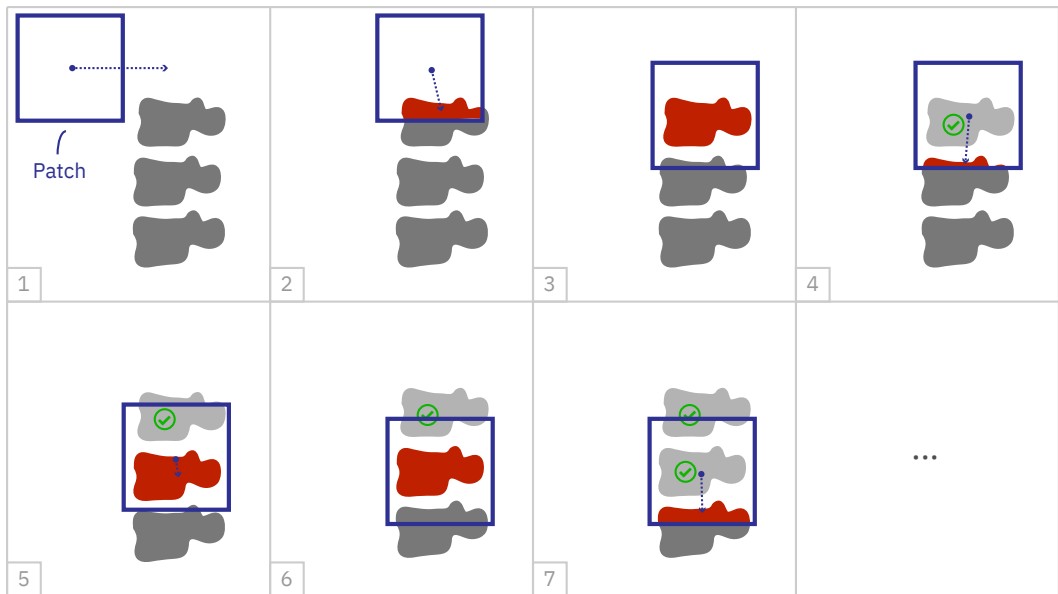

Figure 2: Illustration of the iterative instance segmentation and traversal strategy. The patch is first moving in a sliding window fashion over the image (1), until a fragment of vertebral bone is detected (2). The patch is then moved to the center of the fragment until the entire vertebra becomes visible (3). The segmented vertebra is added to the instance memory and the same patch is analyzed again, now yielding a fragment of the following vertebra – the updated memory forces the network to ignore the previous vertebra (4). The patch is centered now at the detected fragment of the following vertebra and the process repeats (5-7).

position converges because no additional voxels are detected anymore that would affect $\boldsymbol{b}_t$. We detect convergence by comparing the current patch position $\boldsymbol{x}_t$ and the previous patch position $\boldsymbol{x}_{t-1}$, testing whether they still differ by more than $\delta_{\max}$ on any axis. In rare cases, the patch position does not converge but keeps alternating between positions that differ slightly more than the threshold $\delta_{\max}$. We therefore limit the number of iterations per vertebra. When this limit is reached, we assume that the patch has converged to the position between the two previous patch positions and we accordingly move the patch to $\boldsymbol{x}_{t+1} = \left\lceil \left( \boldsymbol{x}_t + \boldsymbol{x}_{t-1} \right) / 2 \right\rceil$. We set $\delta_{\max} = 2$ for our experiments, $\Delta\boldsymbol{x}$ to half the patch size and the maximum number of iterations to 20.

Once the position has converged, the segmented vertebra is added to the output mask using a unique instance label and the segmentation mask obtained at this final position. Furthermore, the instance memory is updated. In the following iteration, the network analyzes the same patch again. The updated memory prompts the network to detect a fragment of the following vertebra and the patch is moved to the center of the detected new fragment, repeating the segmentation process for the next vertebra. The entire process continues until no further fragments are found, i.e., until all visible vertebrae are segmented.

**Classification of vertebra completeness**

An obvious strategy for disregarding incompletely visible vertebrae in the segmentation process would be to train the network only with examples of fully visible vertebrae. However, the traversal scheme requires detection of vertebral fragments in the patches. We therefore include partially visible vertebrae in the training data, but add a classification component to the segmentation network that classifies each segmented vertebra as complete or incomplete. The U-net architecture consists of a compression and a decompression path, which are commonly understood as recognition and segmentation paths, respectively. The classification component is therefore appended to the compression path as further compression steps. Essentially, it further compresses the input patch into a single value, the estimated probability that the vertebra is completely visible. During traversal, vertebrae classified as incomplete are not added to the output mask, but are added to the instance memory to

facilitate further traversal because not only the last but also the first vertebrae are often incompletely visible.

**Training the network**

The patches used to train the network were selected randomly (⅙), or were forced to contain vertebral bone (⅚) by randomly selecting a scan and a vertebra visible in that scan, followed by random patch sampling within the bounding box of that vertebra. Vertebrae were considered completely visible in a patch when they were manually marked as completely visible in the scan and when not more than 3 % of the volume of the vertebra was not contained in the patch. This allows for some tolerance as manual identification of incompletely visible vertebrae can be ambiguous in scans with low resolution.

Due to the size of the input patches, the Nvidia Titan X GPUs with 12 GB memory that we used for training allowed processing of only single patches instead of minibatches of multiple patches. We therefore used Adam [15] for optimization with a fixed learning rate of 0.001 and an increased momentum of 0.99, which stabilizes the gradients. Furthermore, the network predicts labels for all voxels in the input patch and the loss term is accordingly not based on a single output value, but on more than two million output values per patch. The loss term $\mathcal{L}$ combines terms for segmentation and classification error:

$$\mathcal{L} = \underbrace{\lambda \cdot \mathrm{FP}_{\mathrm{soft}} + \mathrm{FN}_{\mathrm{soft}}}_{\text{Segmentation error}} + \underbrace{(-t \log p - (1-t) \log(1-p))}_{\text{Classification error}}$$

We propose to optimize the segmentation by directly minimizing the number of incorrectly labeled voxels, i.e., the number of false positives and false negatives. This is similar to loss terms based on the Dice score [16], but values are more consistent across empty and non-empty patches because the number of true positives is not part of the term. The factor $\lambda$ weights the cost of a false positive error relative to a false negative error, which is useful to counteract an imbalance between background and foreground voxels. In most segmentation problems, the number of background voxels is much larger than the number of foreground voxels and consequently a systematic false negative mistake is much more favorable than a systematic false positive mistake. We used $\lambda = 0.1$ in all experiments.

Given an input patch $I$ and for all voxels $i$ binary reference labels $t_i$ and probabilistic predictions $p_i$, differentiable expressions for the number of false positive and false negative predictions are:

$$\mathrm{FP}_{\mathrm{soft}} = \sum_{i \in I} \omega_i \cdot (1 - t_i)\, p_i \qquad \mathrm{FN}_{\mathrm{soft}} = \sum_{i \in I} \omega_i \cdot t_i\, (1 - p_i)$$

Here, $w_i$ are weights used to assign more importance to the voxels near the surface of the vertebra. This aims at improving the separation of neighboring vertebrae [14]. The weights are derived from the distance $d_i$ of voxel $i$ to the closest point on the surface of the targeted vertebra: $\omega_i = \gamma \cdot \exp\left(-\frac{d_i^2}{\sigma^2}\right) + 1$. We used $\gamma = 8$ and $\sigma = 6$ in all experiments.

The classification error is defined as the binary cross entropy between the true label $t$, which is a binary value that indicates whether the vertebra is completely visible in the patch, and the predicted probability for complete visibility $p$.

During training, the status of the instance memory is derived from the reference segmentation mask. We used random elastic deformations, random Gaussian noise, random Gaussian smoothing as well as random cropping along the z-axis to augment the training data. Dropout with a drop rate of 20 % was used between each pair of consecutive convolutional layers. Rectified linear units were used as activation function in all layers except the output layers, in which sigmoid functions were used. The network was implemented using the PyTorch framework. Training on an Nvidia Titan X GPU took about 3–4 days.

# 3 Experiments

## 3.1 Datasets

We trained and evaluated the method with two different sets of CT scans that visualize the spine. The first set consisted of 15 spine CT scans from the spine segmentation challenge hold in conjunction with the Computational Spine Imaging (CSI) workshop at MICCAI 2014 [6]. The second set consisted of 35 low-dose chest CT scans from the National Lung Screening Trial (NLST), which was a large study in the United States that investigated the use of low-dose chest CT for lung cancer screening [17].

The CSI dataset consists of dedicated spine CT scans that visualize all thoracic and lumbar vertebrae. In the original challenge, 10 scans were available for training and 5 scans were used for evaluation. To allow for a comparison of the results, we used the same division in our experiments. The scanned subjects were healthy young adults (20–34 years). Parameters of the scans and reconstructions were 120 kVp, IV-contrast, an in-plane resolution of 0.31 mm to 0.36 mm and a slice thickness of 0.7 mm to 1.0 mm. Reference segmentations of most of the visible vertebrae were provided by the challenge organizers. These were generated in a semi-automatic fashion by manually correcting automatic segmentations. Because our method detects all vertebrae, even if they are only partially visible in the scan, we cropped all scans to remove vertebrae that were not included in the reference segmentations.

The NLST dataset consists of low-dose chest CT scans that were acquired for lung screening and therefore visualize only a variable section of the thoracic vertebrae and the upper lumbar vertebrae. The scanned subjects were heavy smokers aged 50 to 74 years. Parameters of the scans and reconstructions were 120 kVp or 140 kVp, no IV-contrast, an in-plane resolution of 0.54 mm to 0.82 mm and a slice thickness of 1.0 mm to 2.5 mm. We randomly divided the set into subsets for evaluation of the segmentation performance (test set, 10 scans), for validation during method development (5 scans) and for training (20 scans). In the scans of the test and the validation sets, reference segmentations of all visible vertebrae were obtained manually by drawing along the contour of each vertebra in sagittal slices using an interactive live wire tool [18]. The contours were converted into segmentation masks, in which inaccuracies and other mistakes were corrected voxel-by-voxel. In the scans of the training set, we obtained reference segmentations in a semi-automatic fashion, which requires substantially less manual annotation effort than fully manual segmentation. A preliminary version of the network was trained with the validation scans and used to predict rough segmentations in the training scans, which were manually inspected and corrected voxel-by-voxel and used to train the network.

In both datasets, all vertebrae were manually labeled as either completely or partially contained in the scan. The scans in the CSI dataset all extend beyond the sacrum so that only the upper vertebrae were incomplete.

## 3.2 Evaluation metrics

The segmentation performance was evaluated using the same metrics that were used in the CSI 2014 spine segmentation challenge as this allows for a comparison with the challenge results [6]. These metrics were the Dice coefficient and the mean absolute surface distance (ASD). Both metrics were calculated for individual vertebrae and then averaged over all scans. Each vertebra in the reference segmentation was compared with the vertebra in the automatic segmentation mask with which it had the largest overlap. For this evaluation, all automatically detected vertebrae were considered, even if they were classified as incompletely visible. The classification performance was evaluated using the classification accuracy and the average number of false positives and false negatives per scan.

## 3.3 Results

We trained an independent instance of the network for each of the two datasets. To demonstrate that both directions of traversal are feasible, we trained the network for the CSI dataset to traverse downwards along the spine and the network for the NLST dataset to traverse upwards. The segmentation and classification performance on both datasets is summarized in Table 1. Overall, the network performed slightly better on the dedicated, high resolution spine CT scans than on the low-dose chest CT scans. In the CSI dataset, our method achieved a performance slightly better than that of our previous two-stage approach [13] and that of the best performing method that participated in

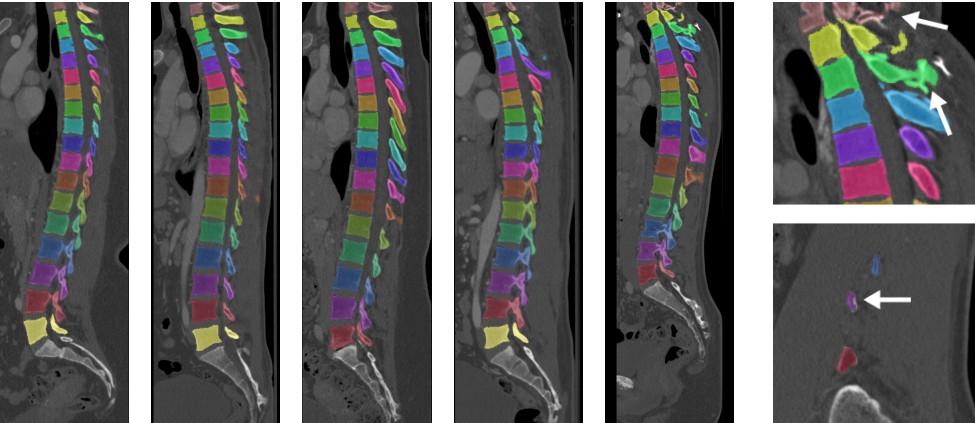

Figure 3: Segmentation results obtained for the spine CT scans in the CSI 2014 challenge dataset. Vertebrae classified as incomplete are shown semi-transparent. The two cutouts on the right illustrate segmentation errors: oversegmentation (top) and minor undersegmentation (bottom).

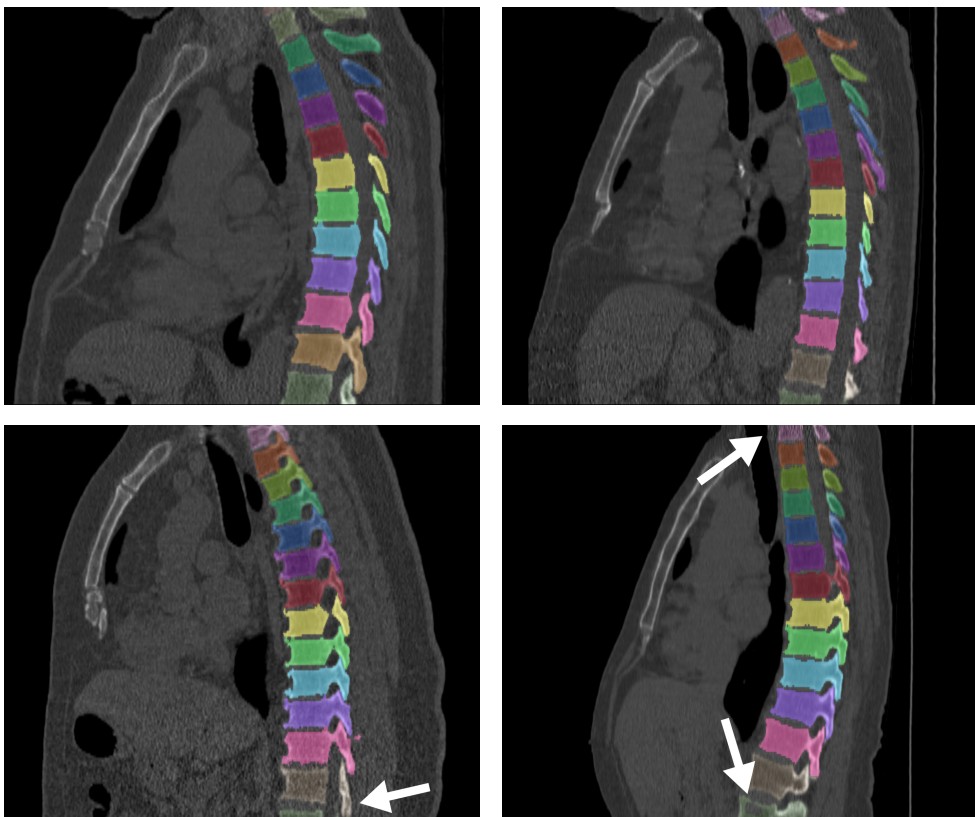

Figure 4: Segmentation results in low-dose chest CT scans. Vertebrae classified as incomplete are shown semi-transparent. The top row shows segmentations with only correct classifications and without larger segmentation errors. The bottom row shows a classification error (left) and a segmentation error (right, bottom arrow). The upper arrow points in the bottom right figure points at an area with strong low-dose imaging artifacts. Note that the vertebra was, however, correctly segmented and classified.

Table 1: Quantitative results of automatic segmentation and completeness classification

| | Segmentation | | Classification | | |
| Dataset | Dice score | ASD | Accuracy | FP / scan | FN / scan |
|---|---|---|---|---|---|
| Spine CT (CSI) | 95.8 % | 0.19 mm | 100 % | 0 | 0 |
|     challenge winner [19] | 94.7 % | 0.37 mm | - | - | - |
|     our previous method [13] | 94.8 % | 0.29 mm | - | - | - |
| Low-dose chest CT (NLST) | 92.1 % | 0.34 mm | 95.7 % | 0 | 0.6 |
|     slice thickness $< 2$ mm | 92.0 % | 0.40 mm | 93.1 % | 0 | 1.0 |
|     slice thickness $\geq 2$ mm | 92.2 % | 0.32 mm | 96.8 % | 0 | 0.4 |

the original challenge [6]. Examples of automatically obtained segmentation masks are shown in Figures 3 and 4.

Most vertebrae were correctly classified as completely or partially visible. Notably, the network made mistakes only in chest CT scans and only in vertebrae close to the beginning or end of the visible section of the spine. There were no mistakes where the network predicted an implausible sequence by labeling a vertebra between two completely visible vertebrae as incompletely visible, or vice versa.

The number of required iteration steps and thus the runtime depends on the size of the image and the number of visible vertebrae. In the spine CT dataset, on average 47 iteration steps were needed at a runtime of 0.5 s per step. The total runtime per scan was therefore on average 24 s for segmentation of 17 or 18 vertebrae, excluding time required for loading the image and storing the results. Methods that participated in the CSI 2014 challenge had a runtime ranging from 10 s per vertebra $\approx 3$ min to 45 min per case. In the chest CT scans, fewer vertebrae were visible, on average 14 per scan. However, since these scans have a larger field of view as they also contain the entire lungs, more iteration steps were needed initially to find the first vertebra, i.e., to locate the spine in the image. The average number of iteration steps was 56 and the average runtime was about 27 s per scan.

## 4   Discussion and Conclusions

In this paper, we demonstrated that a patch based segmentation network is capable of learning a complex instance segmentation task, namely vertebra segmentation in dedicated as well as non-dedicated CT scans. The network was able to learn to infer from an additional memory input which object to segment and was additionally able to perform segmentation and classification tasks concurrently. The proposed method outperformed all other methods for vertebra segmentation that participated in the CSI 2014 spine segmentation challenge [6] and performed well in a challenging set of low-dose chest CT scans.

Accurate and fast vertebra segmentation in low-dose chest CT scans enables the analysis of lung screening exams to incidentally detect early signs of osteoporosis by detecting vertebral compression fractures or by measuring the bone mineral density. In this context, it is noteworthy that our method did not perform substantially worse in low-dose chest CT scans compared to dedicated spine CT scans, even though there are large differences in image quality and variability of the field of view. Segmentation and classification errors in the chest CT scans were most often found in the first and last visible vertebrae, in which the network has less context available. These vertebrae are also often incompletely visible, which makes their segmentation more difficult. However, increasing the number of training scans and thus the number of examples of such difficult instances might lead to further performance improvements. For this, the semi-automatic segmentation scheme used for the chest CT training set could be repeated to efficiently generate more diverse training data.

We used chest CT scans to evaluate the performance on images that were originally not acquired for analysis of the spine. The proposed method is neither restricted to spine and chest CT scans, nor to CT as the imaging modality. The method could be also applied to, e.g., abdominal CT or MR scans that partially cover the spine. An interesting direction for further research would be the training of a single instance of the network using training data with a large variety of fields of view, imaging

modalities and image resolutions and qualities. Also, an evaluation on a very diverse set of images could further support the general applicability of the proposed method.

For some applications, it might be relevant to anatomically label each detected vertebra. Provided that a large enough set of training data would be available, the proposed network could potentially be extended with a forth component that additionally provides this anatomical labeling.

In conclusion, this paper presents an instance segmentation approach to vertebra segmentation that is fast, flexible and eminently accurate both in dedicated scans of the spine as well as in non-dedicated scans containing an arbitrary section of the spine.

### Acknowledgments

We are grateful to the United States National Cancer Institute (NCI) for providing access to NCI's data collected by the National Lung Screening Trial. The statements contained herein are solely ours and do not represent or imply concurrence or endorsement by NCI. We would also like to thank the organizers of the CSI 2014 spine segmentation challenge for making scans and reference segmentations publicly available.

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
