# OpenReview forum: "Iterative fully convolutional neural networks for automatic vertebra segmentation"
_MIDL.amsterdam/2018/Conference — MIDL 2018 Oral_

### Review · AnonReviewer1 · 2018-05-07
**This is an overall good quality manuscript, and the readers may appreciate the iterative instance segmentation idea.**

**Rating:** 4
**Confidence:** 2

**Review:**

This paper presents an iterative CNN framework for automatic vertebra segmentation, which is quite important in quantitative image analysis. The paper is well written and easy to follow, and the methodology is sound and clearly explained. I really like the iterative instance segmentation idea. The experimental result is intact and comprehensive. However, the authors should consider doing some statistics test on the reported results that compared with other methods to see if they are statistically significantly different.

Overall, the topic of this paper is interesting for the MIDL readers and I recommend this paper for acceptance for publication.


**Special Issue:**

Yes

---

### Review · AnonReviewer2 · 2018-05-07
**The authors propose an interesting iterative instance segmentation framework based on FCN for the successive (bottom-up/top-down) segmentation of vertebrae in CT scans. A memory module is integrated into the original U-Net architecture to remember the vertebras (i.e., instances) have been segmented and move forward to segment the following nearest vertebrae. The proposed method has been evaluated on two datasets, while the comparison with the state-of-the-art methods was limited.**

**Rating:** 3
**Confidence:** 2

**Review:**

The authors extend the original U-Net architecture by including a memory module for successive instance segmentation as well as an additional contracting path for the identification of the visibility (i.e., completely vs. partially visible) of each instance.
However, the practical significance of the latter classification task is doubtable. It is unclear if this supplementary task would improve the primary segmentation task. In addition, the evaluation of the performance of this classification task seems to be subjective, since there is no precise definition of “partially” visible.

On the module of instance memory, the authors did not precisely describe how to combine the image path I and the corresponding memory patch M as a two-channel input in Fig. 1. Does it mean I and M are concatenated directly, which are then further processed by the following convolutional blocks?

Also, there is no clue if the propose instance segmentation strategy was superior than the original 3D U-Net or V-Net that segment all vertebras jointly, considering there is no direct comparison in the experimental part. In addition, the instance segmentation strategy perhaps increase the computational complexity during segmentation, since each instance maximally take 20 iterations (controlled by user-defined parameters) to be fully segmented.

The experimental results presented in Sec. 3 is very limited, considering there is no comparison with the state-of-the-art methods (e.g., [8]). The authors only compared their previous method (i.e., [13]) with current method on one dataset.


**Special Issue:**

No

---

### Review · AnonReviewer3 · 2018-05-09
**A very good paper**

**Rating:** 4
**Confidence:** 3

**Review:**

Overall:
The paper proposes a combination of a deep learning model with an iterative procedure for automatic vertebra segmentation. The presented approach uses a segmentation network (U-Net) and a classification network (CNN + dense layer) together with an iterative procedure for finding patches containing vertebrae. The paper is easy to read and all ideas are explained in a lucid manner. The results are fully convincing. The proposed idea (especially the iterative procedure for finding patches of interest) has a great potential to be used in other applications.

Strengths:
+ The paper is very well written and easy to follow.
+ The paper proposes an interesting solution to a vertebra segmentation using deep learning. A novelty of the approach is to combine a segmentation and classification networks with an iterative procedure for selecting patches.
+ The proposed approach could be potentially applied to other problems where prior knowledge about objects of interest could be utilized.
+ The idea of partially sharing segmentation layers and classification layers is clever since there is a deficiency of data.
+ The results are impressive (the best result on a CSI 2014 challenge).
+ The idea of transferring networks between two datasets makes the story more convincing.

Remarks:
* Minor
- Is it possible to introduce further prior knowledge about vertebrae? For instance, about volumes?

**Special Issue:**

Yes

---

### Comment · ~Bram_van_Ginneken1 · 2018-05-18
**Selection for longlist for special issue Medical Image Analysis**

Dear authors,

Congratulations on your acceptance to MIDL! We have selected your paper on the longlist for the Medical Image Analysis Special Issue. Please read this page:
https://midl.amsterdam/special-issue-in-medical-image-analysis/
Please answer the three questions that are listed on that page about your interest in submitting to the special issue, potential overlap with other publications, and related publications.

You can post your answer here directly below on openreview.net, or mail me directly at bram.vanginneken@radboudumc.nl.

Best regards, Bram

---

### Decision · Program_Chairs · 2018-05-15
**Paper87 Acceptance Decision**

Oral